# Association between Plasma HLA-DR+ Placental Vesicles and Preeclampsia: A Pilot Longitudinal Cohort Study

**DOI:** 10.3390/cells13020196

**Published:** 2024-01-20

**Authors:** Marianna Onori, Rita Franco, Donatella Lucchetti, Silvio Tartaglia, Silvia Buongiorno, Giuliana Beneduce, Fabio Sannino, Silvia Baroni, Andrea Urbani, Antonio Lanzone, Giovanni Scambia, Nicoletta Di Simone, Chiara Tersigni

**Affiliations:** 1Faculty of Medicine and Surgery, Catholic University of the Sacred Heart, L.go F.Vito 1, 00168 Rome, Italy; marianna.onori01@icatt.it (M.O.); rita.f7593@gmail.com (R.F.); dnlucchetti@gmail.com (D.L.); fabiosannino2001@gmail.com (F.S.); silvia.baroni@policlinicogemelli.it (S.B.); antonio.lanzone@policlinicogemelli.it (A.L.);; 2Fondazione Policlinico Universitario A. Gemelli IRCCS, L.go A.Gemelli 8, 00168 Rome, Italy; silvio.tartaglia@policlinicogemelli.it (S.T.); silvia.buongiorno@policlinicogemelli.it (S.B.); chiara.tersigni@policlinicogemelli.it (C.T.); 3Department of Biomedical Sciences, Humanitas University, 20072 Milan, Italy; 4IRCCS Humanitas Research Hospital, 20089 Milan, Italy

**Keywords:** preeclampsia, syncytiotrophoblast-derived extracellular vesicles, HLA-DR, biomarker

## Abstract

(1) Background: Preeclampsia (PE) usually presents with hypertension and proteinuria, related to poor placentation. Reduced maternal–fetal immunological tolerance is a possible trigger of inadequate placentation. Aberrant antigen expression of HLA-DR has been observed in the syncytiotrophoblast of PE patients. In this study, we analyzed plasma levels of Human Leukocyte Antigen (HLA)-DR+ syncytiotrophoblast-derived extracellular vesicles (STEVs) during the three trimesters of pregnancy in relation to PE onset. (2) Methods: Pregnant women underwent venous blood sampling during the three trimesters. STEVs were collected from plasma via ultracentrifugation (120,000 g) and characterized by Western blot, nanotracking analysis and flow cytometry for the expression of Placental Alkaline Phosphatase (PLAP), a placental-derived marker, and HLA-DR. (3) Results: Out of 107 women recruited, 10 developed PE. STEVs were detected in all three trimesters of pregnancy with a zenith in the second trimester. A significant difference was found between the non-PE and PE groups in terms of plasma levels of HLA-DR+ STEVs during all three trimesters of pregnancy. (4) Conclusions: More research is needed to investigate HLA-DR+ as a potential early marker of PE.

## 1. Introduction

Preeclampsia (PE) is a common disorder of pregnancy that complicates 2–8% of all pregnancies and remains a major cause of severe maternal and newborn morbidity and mortality worldwide [1].

It usually presents with hypertension and proteinuria generally in the second half of pregnancy, although the clinical presentation can be extremely heterogeneous [2]. Clinically overt PE comes from factors released into the maternal circulation from the placenta as a result of syncytiotrophoblast stress due to defective placentation.

It is the placentation defect, which occurs in the first half of pregnancy, that plays a key role in the pathogenesis of preeclampsia. In fact, according to the “three-stage model,” in the first half of pregnancy, there is inadequate invasion of the trophoblast into the placental bed, causing inadequate remodeling of the utero-placental spiral arteries (stage 2).

Inadequate placental development can become clinically relevant at different gestational ages depending on its severity, leading, in the third stage of the disorder, to utero-placental malperfusion and oxidative stress of the syncytiotrophoblast [3].

The first trigger of the pathogenetic cascade could be the reduced tolerance of the semi-allogeneic fetus by the maternal immune system, which is closely associated with spiral artery remodeling at the beginning of the second trimester [4]. The semi-allogeneic trophoblast forms several interfaces with the maternal immune system [5]. The syncytiotrophoblast forms the epithelial lining of chorionic villi and has a large surface area in contact with the blood of maternal circulating immune cells. Extracellular vesicles derived from the syncytiotrophoblast (STEVs) and shed into the maternal circulation represent the main maternal–fetal interface in the second half of pregnancy.

Successful pregnancy depends on tight control of Human Leukocyte Antigen (HLA) class I and class II expression in the villous (VT) and extravillous trophoblast (EVT). Specifically, the lack of HLA class II molecule (-DP, -DQ and -DR) expression on trophoblasts prevents the immune response of maternal T cells against paternally derived antigens.

Intriguingly, proteomic analysis of extracellular syncytiotrophoblast-derived vesicles (STEVs) isolated from the placenta of women with PE by dual placental perfusion has shown the exclusive presence of the HLA-DR molecule in STEVs from PE cases compared to controls (Tannetta D, unpublished data).

HLA-DR is a class II molecule constitutively expressed on professional antigen-presenting cells (APCs) to present exogenous antigens to T cells to elicit an antigen-specific immune response.

In a previous study, aberrant expression of the HLA-DR antigen was found in approximately 40% of STEVs obtained from PE women by double placenta perfusion and confirmed by immunohistochemistry on placental sections [6]

A subsequent study from the same group demonstrated the abnormal expression of HLA-DR in serum STEVs in women with clinically overt PE (64%) during the third trimester of pregnancy. None of the normal pregnant women showed detectable positivity for HLA-DR in circulating STEVs [7].

Currently, the most validated first-trimester screening method for PE is that developed by the Fetal Medicine Foundation (FMF). It combines maternal characteristics, medical and obstetrical history, physical parameters (mean arterial blood pressure, uterine artery Doppler) and biological markers, namely Placental Growth Factor (PlGF) and pregnancy-associated plasma protein A (PAPP-A). The FMF screening test for PE has a detection rate of 77% for early PE and 40% for late PE, with a false positive rate of 10% [8,9].

Based on the evidence of aberrant expression of HLA-DR in the STEVs in PE, we aimed to test HLA-DR carried by STEVs in maternal blood as a possible marker to be investigated for its predictive value in the early screening of PE.

## 2. Materials and Methods

### 2.1. Patients

This study has been designed and conducted according to the principles of the Declaration of Helsinki and approved by the Ethics Committee of the Fondazione Policlinico A. Gemelli IRCCS, Rome, Italy (ID:4299 date of approval 15 July 2021). Written informed consent was obtained from all recruited individuals. All women enrolled in this study were selected between 11 and 13 weeks +6 days of gestation, among those referred to our Obstetrics Outpatient from September 2022 to September 2023 to undergo first-trimester screening for trisomies and PE, according to the Fetal Medicine Foundation guidelines (https://fetalmedicine.org/ accessed on 30 May 2021). High risk for PE was defined as a result ≥ 1:100. For the identification of cases of PE, we referred to the definition of the International Society for the Study of Hypertension in Pregnancy (ISSHP) [10]. Women aged < 18 years, those unable to give informed consent, and those with infectious diseases and/or twin pregnancies were excluded from this study.

### 2.2. Sample Collection

All women enrolled in this study underwent venous blood sampling by venipuncture from the antecubital fossa at booking (11 and 13 weeks +6 days of gestational age), during the second trimester (19–22 weeks of gestation) and the third trimester (28–32 weeks of gestation). Blood was collected in a VACUETTE^®^ test tube for serum (3 mL) and with sodium citrate for plasma (2 mL). Each sample was centrifuged at 1200 g for 10 min at 20° to remove the cellular components and obtain serum or plasma and then frozen in 500 μL aliquots at −80 °C until further use.

### 2.3. Isolation of STEVs from Plasma

Plasma samples were thawed at R/T. Each sample was diluted (1:1 *v*/*v*) with PBS (Dulbecco’s Phosphate Buffered Saline, Biowest), centrifuged at 3000× *g* for 30 min at 4 °C and then ultracentrifuged at 120,000× *g* for 90 min at 4 °C using Optima TL ultracentrifuge (Beckman Coulter, Brea, CA, USA). After ultracentrifugation, the supernatant was removed, and the pellet, containing extracellular vesicles (EVs) of various sizes, was washed with PBS and ultracentrifuged again at 120,000× *g* for 90 min at 4 °C. Finally, the supernatant was discarded, and the EV pellet was resuspended with 100 μL of PBS. To quantify STEVs, samples were first lysed with 0.25% NP40 for 60 min; the protein concentration was then determined with a Bradford assay.

### 2.4. Western Blot

For the analysis of PLAP expression, STEV samples were isolated from the plasma of 3 patients during the first, second and third trimesters of pregnancy and lysed on ice with NP40 at 0.25% for 60 min. Samples were denatured at 95 °C and then subjected to separation by SDS/PAGE (ThermoFisher Scientific, Waltham, MA, USA). Subsequently, semi-dry transfer was performed on PVDF membranes (GE Healthcare Life Sciences, Cleveland, OH, USA). Non-specific bands were blocked with TBS-T (0.1% tween 20 BioRad Laboratories, DPBS Biowest, Hercules, CA, USA) containing 5% non-fat dry milk (Blotting-Grade Blocker, BioRad Laboratories, Hercules, CA, USA) for one hour at R/T. The membrane was incubated with a murine monoclonal anti-PLAP antibody (NDOG2, University of Oxford, Oxford, UK; 1 μg/mL) in 5% milk solution at 4 °C overnight. PLAP is Placental Alkaline Phosphatase, mainly expressed in human placentae and largely used as a specific marker of syncytiotrophoblast [11].

The membranes were washed in TBS-T before incubation with a horseradish peroxidase-conjugated secondary antibody (diluted 1:5000 in milk solution, Invitrogen, Waltham, MA, USA) for 1 h at R/T.

After washing, the membranes were treated with a chemiluminescence system (PierceTM, Thermo Fischer Scientific, Waltham, MA, USA) and exposed to Hyperfilm ECL (GE Healthcare Life Sciences, Cleveland, OH, USA). Western blot densitometric analysis was performed using Nine Alliance Q9 software (Uvitec Alliance, Cambridge, UK). Protein content normalization was performed with β-actin, a common normalizing control for STEVs [12].

### 2.5. Nanoparticle Tracking Analysis (NTA)

Nanoparticle tracking analysis was performed by NanoSight NS3000 (Malvern Panalytical, Malvern, UK) to visualize EVs by laser light scattering to assess the concentration/size of EVs. EVs were diluted 1:100 or 1:50 with 0.1 μm filtered PBS to specifically fit the optimal working range (20–120 particles/per frame) of the instrument. Five 60 s videos were recorded under the flow mode for each sample with the camera level set at 14 and the detection threshold set at 5. The ratio of total completed tracks/total valid tracks was <10.

Data were analyzed with NTA 3.0 software (NTA 3.4 Build 3.4.4 Malvern Panalytical), which provided high-resolution particle size distribution profiles and EV concentration measurements (n of particles/mL).

### 2.6. Flow Cytometry

The analysis of STEVs was performed by multi-color flow cytometry using a CytoFLEX S cytometer (Beckman Coulter) equipped with violet laser (405 nm) excitation sources. This instrument can record the SSC (side scatter) for the blue laser (BSSC) and the violet laser (VSSC).

STEVs were marked for 30 min at 37 °C with Calcein Violet AM (25 μg desiccate Invitrogen by Thermo Fisher Scientific) to distinguish intact vesicles from cell debris.

Before the STEV samples were run, the 0.22 μm filtered PBS was analyzed three times for 2 min to remove the background (approximately 200 events per second).

To confirm the placental origin of the vesicles, STEVs were incubated for 1 h at 4 °C in the dark with an anti-PLAP (placental anti-alkaline phosphatase) PE (phycoerythrin)-conjugated murine monoclonal antibody (8B6 sc-47691, Santa Cruz, Santa Cruz, CA, USA) or its control IgG1 PE-conjugated (Biolegend UK Ltd., Cambridge, UK). PLAP is exclusively expressed in placental tissue and is commonly used to distinguish vesicles released from the placenta from those derived from other cell types. To analyze the expression of HLA-DR, STEVs were incubated for 1 h at 4 °C in the dark with an anti-HLA-DR FITC-conjugated murine monoclonal antibody (ab1182, Abcam, Cambridge, UK) or its control IgG1 FITC-conjugated antibody (Abcam). To evaluate the exosome percentage of STEVs analyzed, samples were incubated for 1 h at 4 °C in the dark with an anti-CD63 violet-conjugated murine monoclonal antibody (H5C6, Biolegend) or its control IgG1 violet-conjugated antibody (Biolegend). PLAP-positive and HLA-DR-positive vesicles were normalized to positive calcein events. Before use, all antibodies were diluted with 0.22 μm filtered PBS to minimize interference from background particles. Fluorochrome compensation for the multi-colored STEV cytofluorometer was performed using BD Compbeads (BD Biosciences, Franklin Lakes, NJ, USA), labeled with fluorescence-conjugated antibodies. Subsequently, 0.22 μm filtered PBS was added to the labeled samples to reach 500 μL and evaluated immediately in the cytofluorometer (Beckman Coulter). Data were acquired and analyzed by CytExpert 2.2TM software (version 2.2, CytoFLEX S, Beckman Coulter, Milan, Italy).

### 2.7. Placental Growth Factor (PlGF) Analysis

Serum PlGF concentration was measured by an automated device using an ECLIA assay according to the manufacturer’s instructions (Cobas E601, Roche Diagnostics, Rotkreuz, Switzerland). One hundred microliters of serum samples were used for each well.

### 2.8. Statistical Analysis

The clinical data are shown as the mean ± standard deviation (SD) or percentage (%), depending on the type of variable, and were analyzed using Student’s t-test. All data were analyzed for normal distribution using the Shapiro–Wilk test and analyzed by the Mann–Whitney U-test performed with Prism software version 9.0. For all analyses, *p* < 0.05 was considered significant.

## 3. Results

### 3.1. Clinical Results

The clinical characteristics of the study population enrolled in this study are shown in Table 1. The population has been divided into two groups of risk for PE, according to the first-trimester combined screening. Ninety women were assigned to the low-risk group (LR) while seventeen women were assigned to the high-risk group (HR). A significantly higher prevalence of proteinuria, previous history of PE, chronic hypertension, diabetes mellitus type II (DMII) and gestational diabetes mellitus (GDM) was seen in the HR group compared to the LR group. As expected, a higher percentage of women using anti-hypertensive drugs, aspirin and heparin was found in the HR group compared to the LR group. A higher average pulsatility index of uterine arteries, as well as lower levels of Pregnancy-Associated Plasma Protein-A (PAPP-A) were found in the HR group compared to the LR group. No significant differences were found in terms of PlGF between the two groups. A higher prevalence of cases of PE was observed in the HR group compared to the LR group, as well as lower gestational age at delivery and neonatal birth weight.

Among the study population of 107 women, 10 developed PE (9.3%). In particular, seven patients developed late PE (>34 weeks of gestation) (70%) and three women developed early PE (<34 weeks of gestation) (30%). Nine patients with PE (six late and three early cases) had been previously assigned to the high-risk group (HR) and one (late PE) to the low-risk group (LR), accordingly to the FMF first-trimester screening.

A significant difference in terms of substantial proteinuria; chronic hypertension; greater use of anti-hypertensive drugs, aspirin and heparin; higher MAP; lower levels of PAPP-A; and lower gestational age at delivery, neonatal birth weight and percentile was observed in the PE group compared to the non-PE group (see Table 2).

### 3.2. STEVs Can Be Detected in Peripheral Blood of Pregnant Women from Early First Trimester of Pregnancy

A significant presence of PLAP+ STEVs was detected in the plasma of all women recruited in this study during the three trimesters of pregnancy by Western blot (Figure 1a,b). Nanotracking analysis of STEVs showed a population of STEVs ranging from 100 to 300 nm, with a mean size of 162 nm and a modal size of 113 nm (Figure 1c–e).

Cytometric analysis of PLAP+ EVs showed a significant proportion of syncytiotrophoblast-derived EVs (STEVs) during the first (17.4%), second (19.6%) and third (15.2%) trimesters of pregnancy. In particular, the highest concentration of circulating STEVs was detected during the second trimester with a reduction during the third trimester back to first-trimester levels (Figure 2a). The analysis of CD63 expression of PLAP+ EVs (or STEVs) showed an exosomal population (<100 nm) of only 1% (Figure 2b,c), suggesting that the STEV population analyzed was mainly represented by large STEVs, consistent with NTA results.

### 3.3. Circulating HLA-DR+ STEVs Can Be Detected in Pregnant Women during the Three Trimesters of Pregnancy

HLA-DR+ STEVs were detected in the study population from the first to third trimester of pregnancy (Figure 3a–e). Most cases of HLA-DR-positive STEVs were detected during the first and second trimesters of pregnancy, with a significant loss of positivity during the third trimester (Figure 3d). Within the PE group (*n* = 10), we found six cases (60%) of HLA-DR positivity in the first trimester, six cases (60%) of HLA-DR positivity in the second trimester and one case (10%) in the third trimester. Among the non-PE group (*n* = 97), we found ten cases (10%) of HLA-DR positivity in the first trimester, six cases (6%) in the second trimester and no cases in the third trimester. A significantly higher percentage of HLA-DR+ STEVs was detected in all three trimesters in women who subsequently developed PE compared to controls (non-PE) (*p* < 0.001, *p* < 0.001, *p* < 0.05, respectively) (Figure 3e).

A significant difference in terms of PE prevalence was observed in the HLA-DR+ group compared to the HLA-DR- group (see Table 3).

### 3.4. Levels of HLA-DR+ STEVs and PlGF in PE

Out of sixteen patients with HLA-DR-positive STEVs in the first trimester, six developed PE (38%), with a prevalence of STEV HLA-DR positivity in PE cases, consistent with previous observations [6,7]. A significant difference was found between the non-PE and PE groups in terms of HLA-DR positivity of STEVs during the first trimester of pregnancy (Figure 4a). Sub-analysis for early and late cases of PE confirmed a statistically significant association between HLA-DR positivity and late PE, compared to non-PE (Figure 4b). No association between HLA-DR-positive STEVs and early PE was observed, likely due to the small number of cases observed in our population. The analysis of serum PlGF levels between non-PE and PE showed no significant differences in terms of PlGF levels between non-PE and PE cases (Figure 4c). No significant association was found in the sub-analysis for early and late cases of PE in terms of serum PIGF levels (Figure 4d).

## 4. Discussion

Circulating syncytiotrophoblast-derived extracellular vesicles (STEVs) can be isolated and characterized in pregnancy from the early first trimester. STEVs are widely recognized to be conveyors of biological messages between the mother and the fetus and represent a liquid biopsy of the placenta. STEVs can be used to investigate placental health status since they are carriers of early biomarkers of poor placentation [13]. A previous study from this group showed aberrant expression of HLA-DR molecules in placental sections and STEVs obtained by dual placental perfusion from third-trimester placentae of PE women [6]. In a more recent study, abnormal expression of HLA-DR was found in circulating STEVs isolated by ultracentrifugation from the serum of patients with a clinical diagnosis of PE during the third trimester of pregnancy [7].

In this study, we demonstrated that levels of PLAP+ STEVs increase from the first to the second trimester, likely due to the increase in placental size from the first to the second trimester, then decrease from the second to the third trimester. This observation is consistent with a recent study analyzing circulating STEVs from the sixth to twelfth weeks of gestation [14] but does not confirm a previous observation made in a small population of pregnant women suggesting a linear increase in circulating STEVs from the first to third trimester [15]. Such a discrepancy might be related to the different purification methods used to isolate EVs and thus to the possibly different populations of EVs obtained in terms of size and content. Indeed, in our study and in that by Jiang and co-authors, the purification method for EVs was only ultracentrifugation [14], while in other reports, the purification methods were both ultracentrifugation and concentration gradient [15,16]. Further studies are required to assess whether the two different methods of EV isolation are equivalent or add to the collection of different EV populations. From a translational point of view, what is relevant is that the detection of STEVs in peripheral blood during the first trimester clears a path to a placental liquid biopsy early in pregnancy.

Abnormal suppression of HLA class II molecules might be involved in the immunological pathogenesis of PE. In this study, we aimed to investigate a specific association between abnormal expression of HLA-DR on circulating STEVs and the development of PE. To produce preliminary evidence of the potential value of HLA-DR as a specific marker of PE, we analyzed all women at high risk for PE (*n* = 17) from our study population and some of the low-risk population (*n* = 90) from those recruited during the first-trimester screening in a longitudinal consecutive fashion. Recruitment of patients for this study is still ongoing. This selection bias determined a relatively higher prevalence of PE (9%) than expected (3–4% in Italy). This study bias will be overcome after the analysis of all the population recruited in our study at the end of enrollment. Our preliminary data suggest significantly higher plasma levels of HLA-DR+ STEVs during the three trimesters of pregnancy in women developing PE compared to non-PE controls. The association between HLA-DR positivity and PE suggests the potentiality of this protein as an early biomarker of PE. The authors are aware that the small number of PE cases analyzed in this study does not allow us to draw any conclusions about the predictive value of circulating HLA-DR+ STEVs analyzed in the first trimester of later PE occurrence. This represents a pilot study and further research is needed to evaluate the predictive value of HLA-DR+ STEVs in the screening of PE in a larger population.

## Figures and Tables

**Figure 1 cells-13-00196-f001:**
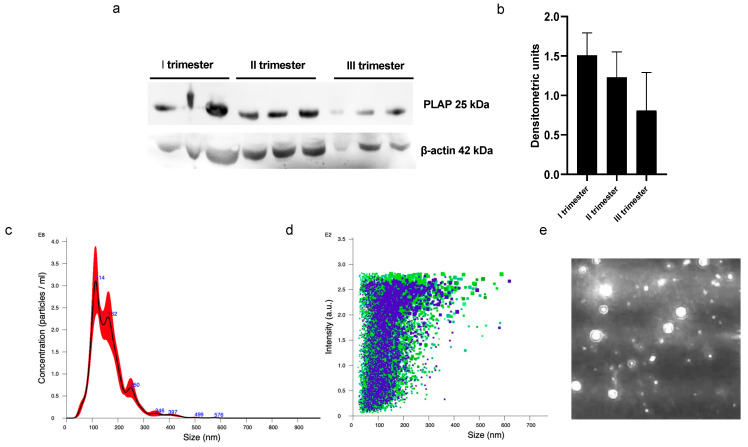
Western blot and NTA analysis of circulating PLAP+ STEVs. (**a**) Western blot analysis showed significant levels of STEVs (PLAP+ EVs) collected from plasma during first (I), second (II) and third (III) trimesters of pregnancy in 3 non-PE cases. (**b**) Densitometric analysis of WB results showed a trend of reduction in circulating levels of STEVs from first to third trimester of pregnancy. (**c**–**e**) Representative experiment of NTA analysis of plasma-derived STEVs during first trimester of pregnancy in a non-PE case, showing indicative concentration (**c**) (black line: average; red area: standard deviation) and intensity of size distribution (**d**) (purple and green refer to different videos) of STEVs. (**e**) Representative image showing placental vesicles at NTA analysis in a non-PE case. STEVs: syncytiotrophoblast-derived extracellular vesicles; PLAP: Placental Alkaline Phosphatase; EVs: Extracellular Vesicles; WB: Western blot; NTA: nanotracking analysis.

**Figure 2 cells-13-00196-f002:**
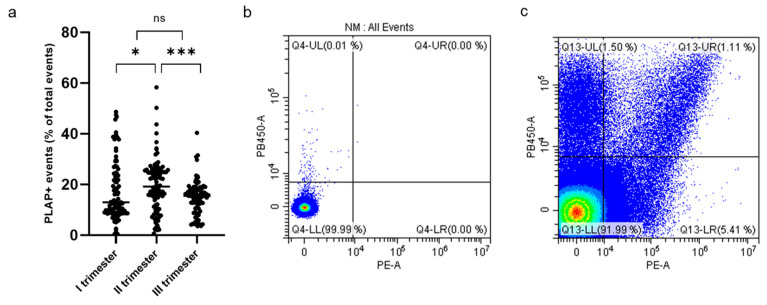
Percentage of PLAP+ STEVs during the 3 trimesters of pregnancy in the study cohort. (**a**) Scatter graph showing concentration of circulating PLAP+ STEVs (% of all circulating EVs) during the 3 trimesters of pregnancy in the study cohort (*n* = 107). A significant increase in STEVs was observed from first to second trimester of pregnancy with a lowering of circulating STEVs during the third trimester. Results are expressed as median. * *p* < 0.05; *** *p* < 0.001; ns not significant. (**b**,**c**) Representative results of flow cytometric analysis of STEVs positive for PLAP (PE) and CD63 (Violet PB450) in a case of late PE during the first trimester of pregnancy (**c**); (**b**) non-marked control. STEVs: syncytiotrophoblast-derived extracellular vesicles; PLAP: Placental Alkaline Phosphatase; EVs: extracellular vesicles.

**Figure 3 cells-13-00196-f003:**
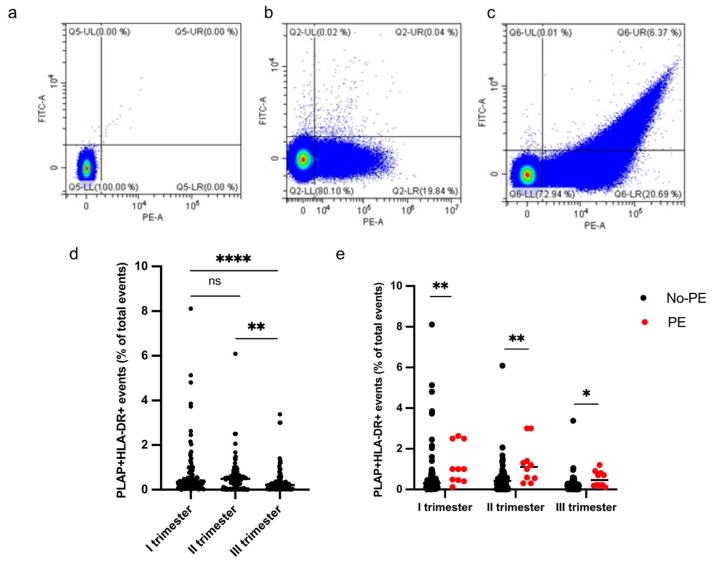
Percentage of HLA-DR+ STEVs during the 3 trimesters of pregnancy. (**a**–**c**) Representative results of flow cytometric analysis of STEVs for PLAP (PE) and HLA-DR (FITC) positivity in a non-PE negative (**b**) and PE positive (**c**) case during the first trimester of pregnancy; (**a**) non-marked control. (**d**) Scatter graph showing concentration of circulating PLAP+HLA-DR+ STEVs during the 3 trimesters of pregnancy in the study cohort (*n* = 107). (**e**) Graph showing levels of circulating HLA-DR+ STEVs during the 3 trimesters of pregnancy in non-PE (*n* = 97) and PE (*n* = 10) groups. Results are expressed as median. Ns: not significant; * *p* < 0.05; ** *p* < 0.01; **** *p* < 0.0001. STEVs: syncytiotrophoblast-derived extracellular vesicles; PLAP: Placental Alkaline Phosphatase; HLA: Human Leukocyte Antigen.

**Figure 4 cells-13-00196-f004:**
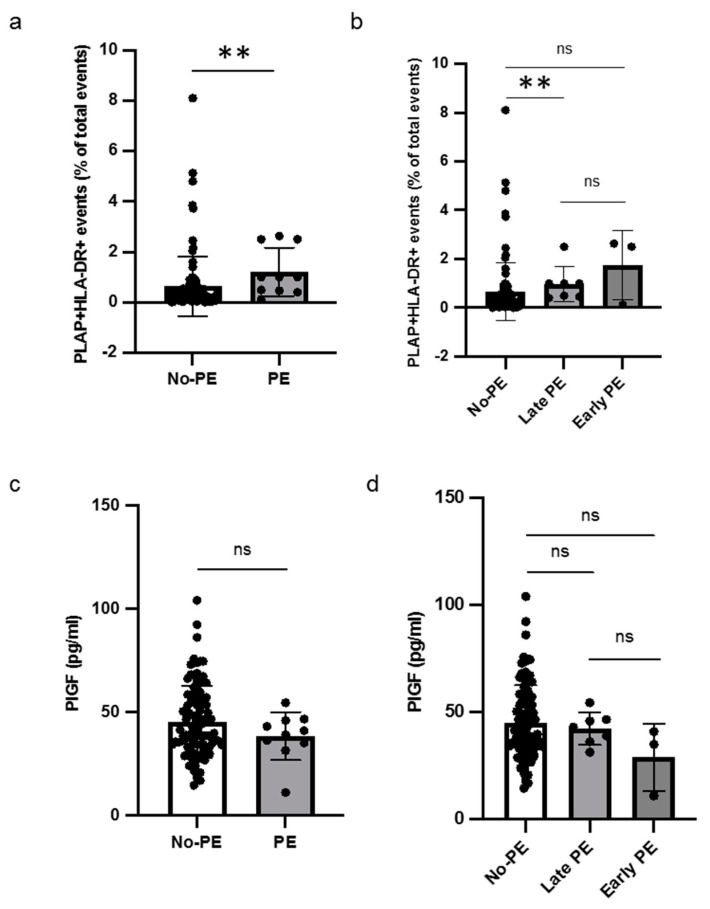
Levels of HLA-DR+ STEVs and PlGF in non-PE and PE women. (**a**,**b**) Scatter graph showing concentration of circulating HLA-DR+ STEVs (% of all circulating EVs) during the first trimester of pregnancy. A significant increase of HLA-DR+ STEVs was observed in PE group compared to non-PE group. (**c**,**d**) Scatter graph showing concentration of PIGF (pg/ml) during the first trimester of pregnancy. No significant differences in terms of PlGF levels between non-PE and PE cases were found. Results are expressed as median. ns: not significant; ** *p* < 0.01. PlGF: Placental Growth Factor; HLA: Human Leukocyte Antigen; STEVs: syncytiotrophoblast-derived extracellular vesicles; EVs: extracellular vesicles; PE: preeclampsia.

**Table 1 cells-13-00196-t001:** Clinical characteristics of the study population according to risk for PE at the first-trimester combined screening.

	Low Risk (*n* = 90)	High Risk (*n* = 17)	*p*
Age (years)	33.5 ± 4.5	34.25 ± 4.4	0.4
BMI (Kg/m^2^)	23.1 ± 11.8	24.7 ± 11.1	0.6
Race			
*White*	84 (93.4%)	16 (94%)	
*Black*	2 (2.2%)	1 (6%)	
*South Asian*	3 (3.3%)	0	
*East Asian*	1 (1.1%)	0	
Proteinuria (%)	1(1.1%)	2 (12%)	<0.05
Medical history			
*Previous PE*	0	2 (12%)	<0.0001
*Family history of PE*	4 (4.44%)	0	0.5
*Chronic hypertension*	0	4 (23.5%)	<0.0001
*DM Type I*	1 (1.1%)	1 (6%)	0.2
*DM Type II*	1 (1.1%)	2 (11.8%)	<0.05
*GDM*	11 (12.2%)	4 (23.5%)	<0.05
*SLE/APS/RA*	3 (3.3%)	1 (6%)	0.7
Cigarette smoking (%)	2 (2.2%)	2(12%)	0.09
Method of conception			
*Spontaneous*	89 (98.8%)	16 (94%)	0.2
*ART*	1 (1.1%)	1 (6%)	0.2
Nulliparous	59 (65.5%)	10 (59%)	0.2
Anti-hypertensive drugs			
*Labetalol*	1 (1.1%)	3 (17.6%)	<0.01
*Metildopa*	1 (1.1%)	5 (29%)	<0.001
*Nifedipina*	0	1 (6%)	<0.01
Aspirin	11 (12.2%)	14 (82%)	<0.0001
Heparin	3 (3.3%)	3 (17.6%)	<0.05
Biomarker			
*MAP* (*mmHg*)	98.4 ± 12.5	105.2 ± 11.5	<0.01
*UtA-PI*	1.6 ± 0.6	2.1 ± 0.5	<0.01
*PlGF* (*pg/mL*)	45.8 ± 17.1	39.0 ± 19.0	0.2
*PAPP-A* (*IU/L*)	2.9 ± 1.6	1.7 ± 0.9	<0.05
PE	1 (1.1%)	9 (53%)	<0.0001
*Early* (<34 weeks)	0	3 (33%)	<0.01
*Late* (>34 weeks)	1 (1.1%)	6 (66%)	<0.01
GA delivery (weeks)	39.1 ± 2.0	36.2 ± 5.5	<0.0001
Neonatal Weight (g)	3217.6 ± 601.6	2684.8 ± 833.4	<0.0001
Percentile (°)	44.6 ± 26.7	23.1 ± 21.4	<0.001

BMI: Body Mass Index; PE: preeclampsia; DM: diabetes mellitus; GDM: gestational diabetes mellitus; SLE: systemic lupus erythematosus; APS: anti-phospholipid syndrome; RA: rheumatoid arthritis; ART: assisted reproductive techniques; MAP: mean arterial pressure; UtA-PI: uterine artery pulsatility index; PlGF: Placental Growth Factor; PAPP-A: Pregnancy-Associated Plasma Protein-A; GA: gestational age.

**Table 2 cells-13-00196-t002:** Clinical characteristics of normal pregnancies (non-PE) versus pre-eclamptic women (PE).

	Non-PE (*n* = 97)	PE (*n* = 10)	*p*
Age (years)	33.27 ± 4.34	36.3 ± 7.89	0.3
BMI (Kg/m^2^)	23.30 ± 11.81	24 ± 12	0.5
Race			
White	91 (93.9%)	9 (90%)	
Black	1 (1%)	1 (10%)	
South Asian	4 (4.1%)	0	
East Asian	1 (1%)	0	
Proteinuria (%)	0	3 (30%)	<0.01
Medical history			
Previous PE	2 (2.1%)	0	0.8
Family history of PE	4 (4.1%)	0	0.7
Chronic hypertension	2 (2.1%)	2 (20%)	<0.05
DM Type I	2 (2.1%)	0	0.8
DM Type II	3 (3.1%)	0	0.7
GDM	14 (14.4%)	1 (10%)	0.5
SLE/APS/RA	3 (3.1%)	1 (10%)	0.4
Cigarette smoking (%)	3 (3.1%)	1 (10%)	0.4
Method of conception			
Spontaneous	96 (98.9%)	9 (90%)	<0.05
ART	1 (1.03%)	1 (10%)	0.09
Nulliparous	64 (66%)	5 (50%)	0.06
Anti-hypertensive drugs			
Labetalol	0	4 (40%)	<0.0001
Metildopa	2 (2.1%)	4 (40%)	<0.0001
Nifedipina	0	1 (10%)	
Aspirin	14 (14.4%)	8 (80%)	<0.0001
Heparin	3 (3.1%)	3 (30%)	<0.01
Biomarker			
MAP (mmHg)	97.06 ± 10.08	119.75 ± 10.34	<0.0001
UtA-PI	1.69 ± 0.56	1.82 ± 0.84	0.6
PlGF (pg/mL)	45.31 ± 17.33	38.65 ± 19.07	0.4
PAPP-A (IU/L)	2.79 ± 1.60	1.12 ± 0.49	<0.01
GA delivery (weeks)	38.96 ± 2.61	34.9 ± 3.98	<0.0001
Neonatal weight (g)	3223.33 ± 575.49	2108.75 ± 1071.8	<0.0001
Percentile (°)	44.64 ± 25.99	8.5 ± 15.94	<0.01

BMI: Body Mass Index; PE: preeclampsia; DM: diabetes mellitus; GDM: gestational diabetes mellitus; SLE: systemic lupus erythematosus; APS: anti-phospholipid syndrome; RA: rheumatoid arthritis; ART: assisted reproductive techniques; MAP: mean arterial pressure; UtA-PI: uterine artery pulsatility index; PlGF: Placental Growth Factor; PAPP-A: Pregnancy-Associated Plasma Protein-A; GA: gestational age.

**Table 3 cells-13-00196-t003:** Clinical characteristics and obstetric outcomes of women according to HLA-DR positivity of STEVs during the first trimester of pregnancy.

	HLA-DR- (*n* = 91)	HLA-DR+ (*n* = 16)	*p*
Age (years)	33.6 ± 4.3	31.6 ± 5.6	0.2
BMI (Kg/m^2^)	23.1 ± 5.0	23 ± 5.4	0.5
Race			
White	87 (95.6%)	16 (100%)	
Black	2 (2.2%)	0	
South Asian	1 (1.1%)	0	
East Asian	1 (1.1%)	0	
Proteinuria (%)	2 (2.2%)	1 (6.3%)	0.4
Medical history			
Previous PE	2 (2.2%)	0	0.6
Family history of PE	3 (3.3%)	1 (6.3%)	0.5
Chronic hypertension	3 (3.3%)	1 (6.3%)	0.5
DM Type I	2 (2.2%)	0	0.6
DM Type II	3 (3.3%)	0	0.5
GDM	15 (16.5%)	0	0.1
SLE/APS/RA	3 (3.3%)	1 (6.3%)	0.5
Cigarette smoking (%)	4 (4.4%)	0	0.5
Method of conception			
Spontaneous	90 (98.9%)	15 (93.7%)	0.1
ART	1 (1.1%)	1 (6.3%)	0.1
Nulliparous	60 (65.9%)	9 (56%)	0.6
Anti-hypertensive drugs			
Labetalol	2 (2.2%)	2 (12.5%)	0.05
Metildopa	5 (5.5%)	1(6.3%)	0.9
Nifedipina	0	1 (6.3%)	
Aspirin	21 (23%)	4 (25%)	0.8
Heparin	5 (5.5%)	1 (6.3%)	0.9
Biomarker			
MAP (mmHg)	97.4 ± 9.2	106.1 ± 20.4	0.1
UtA-PI	1.7 ± 0.6	1.6 ± 0.5	0.4
PlGF (pg/mL)	45.1± 17.4	44.8 ± 17.8	0.9
PAPP-A (IU/L)	2.7 ± 1.6	2.6 ± 1.7	0.7
PE	4 (4.4%)	6 (37.5%)	<0.0001
Early (<34 weeks)	1 (25%)	2 (33.3%)	0.7
Late (>34 weeks)	3 (75%)	4 (66.6%)	0.7
GA delivery (weeks)	38.7 ± 2.7	38.8 ± 3.0	0.8
Neonatal weight (g)	3152.4 ± 594.9	3102.7 ± 888.8	0.6
Percentile (°)	41.9 ± 26.4	40.3 ± 29.4	0.8

BMI: Body Mass Index; PE: preeclampsia; DM: diabetes mellitus; GDM: gestational diabetes mellitus; SLE: systemic lupus erythematosus; APS: anti-phospholipid syndrome; RA: rheumatoid arthritis; ART: assisted reproductive techniques; MAP: mean arterial pressure; UtA-PI: uterine artery pulsatility index; PlGF: Placental Growth Factor; PAPP-A: Pregnancy-Associated Plasma Protein-A; GA: gestational age.

## Data Availability

Data are unavailable due to privacy.

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
