# Peer review of "Association between Plasma HLA-DR+ Placental Vesicles and Preeclampsia: A Pilot Longitudinal Cohort Study"

_cells, 2024, doi:10.3390/cells13020196_

Round 1
Reviewer 1 Report
Comments and Suggestions for Authors
This paper showed that HLA-DR+ placental extracellular vesicles in plasma may be a potential predictive marker for early onset preeclampsia. This is potentially interesting paper. The introduction was as written based on the previous literature. The methods are sufficient and suitable to substantiate authors’ claims. However, I assessed that this article should be rejected for the following reasons.
The number of cases of PE was extremely low, especially with only one case of early PE. It is difficult to draw conclusions based on this number of cases. It is necessary to increase the number of early PE and reconsider. Discussion content was not sufficient. It is necessary to explain why the present results differ from previous reports, including differences in purification methods and study methods of EVs. There was no mention of limitations.
Reviewer 2 Report
Comments and Suggestions for Authors
Onori et al., assessed circulating levels of HLA-DR+ placental EVs in pregnant women during the course of pregnancy. While they show no differences in HLA-DR+ EVs between healthy controls and preeclamptic patients, HLA-DR+ EVs appear to be associated with early onset PE.
The major concern here is that 1 patient had early onset and only 3 others develop PE our of 101 patients. This number is very small for a valid consideration of HLA DR as a potential marker
However, the paper is well written and does show changes in detection of STEVs throughout pregnancy in all populations studied. It also nicely describes the methods and could be one for replicating such a study in a population that has a greater number of high risk patients. Consider changing the title and the emphasis on the biomarker conclusion with such a small number developing PE.
1. As per lines 207-208, authors appeared to have assessed CD63 expression (figure 1a-b). However, no such data was presented in the paper. Please check.
Comment based on CD63 marker-
If you are loading the equal amount of exosomal protein on the gel and the CD63 is present on all exosomes, inferring the amount of EVs from the amount of CD63 is not an appropriate method.
2. Is Beta-actin a good normalizing control for EVs? You can consider normalizing to total protein by Coomassie blue or using stain free gels.
3. In figure 3 along with figure e, I suggest authors present data of HLA-DR positive EVs between non-PE and PE patients during all semesters. That would be more informative than low risk vs high risk.
4. PlGF comes out of the blue. How is this related to EVs? and if there was a reason you assessed it in this study, please discuss.
PLAP expression was measured with westerns but no idea why this was chosen until the last sentence, possibly explain this earlier in the paragraph.
Reviewer 3 Report
Comments and Suggestions for Authors
The authors have chosen a very good and interesting topic. They have showed the association between STEVs HLA-DR+ and early PE, however, no association was observed between HLA-DR+ STEVs and PE. They have also used good methods to elucidate their findings. But as per my understanding, the authors should take following points into consideration to make the article more comprehensive and lucid:
1. Please give reason to use only HLA-DR, as there are other immuno-regulatory markers also released from syncytiotrophoblast (T-cells subtypes, Nk Cells) during dysregulated implantation.
2. A similar study was conducted by Tersigni C et al, 2021. How is this study different/unique from the previous study?
[Tersigni C, Lucchetti D, Franco R, Colella F, Neri C, Crispino L, Sgambato A, Lanzone A, Scambia G, Vatish M, Di Simone N. Circulating Placental Vesicles Carry HLA-DR in Pre-Eclampsia: A New Potential Marker of the Syndrome. Front Immunol. 2021 Sep 3;12:717879.]
3. Results: Section 3.2
"Cytometric analysis of PLAP+ EVs showed a significant proportion of syncytiotrophoblast-derived EVs (STEVs) during the first (13.0%) second (19.2%) and third (15.5%) trimester of pregnancy."
Section 3.3:
"Most cases of HLA-DR positive STEVs were detected during the first and second trimesters of pregnancy with a significant loss of positivity during the third trimester".
These two statements look contradictory.
Comments on the Quality of English Language
1. Many sentences used in the text are little bit longer. Therefore, the authors should go through the text again and try to succinct the long sentences.
2. The writing style needs significant improvement. Please check for grammatical and contextual errors. Rephrasing of sentences would improve the overall quality of the manuscript.
Reviewer 4 Report
Comments and Suggestions for Authors
Dear Authors,
Your study is original and allows us to consider the prospects for using HLA-DR+ in exosomes as an early marker candidate for preclinical diagnosis of PE. In addition, judging by the authors' previous works, aberrant expression of HLA-DR was discovered and confirmed, both in the placenta of pregnant women with PE and in circulating vesicles at the time of delivery. However, this work, in my opinion, is good in concept, but is not sufficient to prove the proposed marker. The sample of patients in this study does not allow us to draw an unambiguous conclusion about the use of HLA-DR in the early stages as a prognostic marker. The number of patients with PE detected from the total pool (101 patients) is too small and the PE cohorts needs to be increased. In addition, it will be possible to build a prognostic model on a larger number of patients.
Overall, the manuscript is well presented. Contains sufficient methods for testing and identifying experimental objects. There are several points that caught our attention.
In the Abstract: the phrase at the end is not clear – “No association was observed between HLA-DR+ STEVs and PE. The presence of HLA- 25 DR+ STEVs in I trimester was associated with early PE onset.” If I understand correctly, these two phrases are mutually exclusive. Clarify, please.
There is also a question regarding the original blots presented. I don't quite understand why the 60 kDa PLAP size is listed at the bottom of the membrane? As far as I understand, this is the bottom of the membrane, and, therefore, there is an area of low molecular weight proteins. Clarify, please.
Round 2
Reviewer 1 Report
Comments and Suggestions for Authors
This paper showed that HLA-DR+ placental extracellular vesicles in plasma may be a potential predictive marker for early onset preeclampsia. This is potentially interesting paper.
This article got better than before, but the authors should change in the following main points and minor points.
Main points
It should be specified whether the analyses shown below were analyses of normal pregnant women (non-PE): western blot and NTA analysis (Figure 1), circulating PLAP+STEVs during the 3rd trimester of pregnancy (Figure 2), and flow cytometoric analysis of STEVs for PLAP and HLA-DR+ (Figure 3 a-d).
Discussion content was not sufficient. It is necessary to explain why this study results differ from previous reports, including differences in purification methods and study methods of EVs. For example, the purification method for EVs were only ultracentrifugation in this study and the article of reference 14, but the purification method were ultracentrifugation and concentration gradient methods in the article of reference 15 paper and Salomon C, et al. (J Clin Endocriol Metab 2017; 102: 3182-3194). The results may differ between these articles because the size of the purified EVs may be different and the contents contained in the EVs may be different. The authors should state the limitations more clearly.
Minor points
Page 1, line 16 in Abstract and Page 2, line56 in Introduction; the Authors should initially use the word ”Human leukocyte antigen” in full, followed by the abbreviation “HLA” in parentheses.
Page 5, line3 in Results; If proteinuria means present rather than previous history, "proteinuria" should be written before "previous history of PE.” Given the current order of description, it is easy to misread this as a history of proteinuria.
Page 8, in Figure 1; Abbreviations of “NTA”, “STEVs”, “PLAP”, “EVs”, and “WB” should be specified in legend.
Page 9, in Figure 2; Abbreviations of “PLAP”, “STEVs”, and “EVs” should be specified in legend.
Page 9, in Figure 3; Abbreviations of “STEVs”, “PLAP”, and “HLA” should be specified in legend.
Page 12, in Figure 4; Abbreviations of “PlGF”, “HLA”, “STEVs”, “EVs”, and “PE” should be specified in legend.
Reviewer 4 Report
Comments and Suggestions for Authors
I have no any comments
Author Response
We thank Reviewer 4 for his/her kind reply.